# Systematic Review and Meta-Analysis of Acute Mortality and Complication Rates Following Leadless Pacemaker Placement Using National-Level Data

**DOI:** 10.3390/medicina61060974

**Published:** 2025-05-25

**Authors:** Akmoldir Sarsenbayeva, Adil Baimbetov, Aras Puodziukynas, Bolatbek Baimakhanov, Alexander Sapunov, Kenzhebek Bizhanov

**Affiliations:** 1Department of Interventional Cardiology, Arrhythmology and Endovascular Surgery, Syzganov National Scientific Center of Surgery, Almaty 050004, Kazakhstan; akmoldir.sarsenbayeva@gmail.com (A.S.); kazpace@gmail.com (A.B.); alex.evasc@gmail.com (A.S.); 2Department of Cardiology, Asfendiyarov Kazakh National Medical University, Almaty 050012, Kazakhstan; 3Hospital of Lithuanian University of Health Sciences “Kauno klinikos”, LT-50161 Kaunas, Lithuania; aras.puodziukynas@gmail.com; 4Department of General Surgery, Syzganov National Scientific Center of Surgery, Almaty 050004, Kazakhstan; bolat.baimakhanov@gmail.com; 5Department of Internal Medicine, Asfendiyarov Kazakh National Medical University, Almaty 050012, Kazakhstan; 6Faculty of Medicine and Healthcare, Al-Farabi Kazakh National University, Almaty 050040, Kazakhstan

**Keywords:** leadless pacemakers, bradyarrhythmia, heart failure, atrial fibrillation, cardiac pacing, transvenous pacemakers, cardiac arrhythmia management, pacemaker complications, pacemaker outcomes, mortality

## Abstract

*Background and Objectives*: Leadless pacemakers provide an innovative alternative to traditional transvenous pacemakers for managing cardiac arrhythmias. The objective of this systematic review is to conduct a meta-analysis comparing acute complication and mortality rates associated with leadless pacemakers versus transvenous pacemaker placements using national-level data. Specifically, we aim to summarize the current evidence and calculate pooled odds ratios for acute overall complications, acute device-related complications, and acute mortality to assess the early safety outcomes of leadless pacemaker placement relative to traditional transvenous pacemakers. *Materials and Methods*: A systematic search of PubMed, Scopus, ScienceDirect, and Google Scholar was conducted by two independent researchers using a predefined search protocol. The search included articles published up to 10 October 2024, without limits on review depth. Studies were included if they provided national-level data comparing leadless pacemaker and traditional pacemaker recipients in terms of acute mortality, acute overall complications, and acute device-related complications. Outcomes were pooled to calculate odds ratios using a random-effects model in RStudio (version 2024.12.1+563). *Results*: A total of five studies met the eligibility criteria. The pooled odds ratio for acute mortality was 2.03 (95% CI: 0.65–6.34, I^2^ = 99%; *p* < 0.01), for acute overall complications was 1.08 (95% CI: 0.45–2.61, I^2^ = 99%; *p* < 0.01), and for acute device-related complications was 1.02 (95% CI: 0.23–4.44, I^2^ = 99%; *p* < 0.01). *Conclusions*: The reviewed studies suggest that leadless pacemakers offer a promising alternative to transvenous pacemakers, offering a comparable short-term safety profile. Ongoing technological advancements may further enhance their applicability in clinical practice.

## 1. Introduction

The development of cardiac pacemakers has become a transformative advance in managing cardiac arrhythmias and heart failure [1]. Bradyarrhythmia is a condition characterized by an abnormally slow heart rate, which can lead to fatigue, dizziness, fainting, and even sudden cardiac death [2]. Pacing strategies have been developed for patients with bradycardia, as well as for those without bradycardia indications but who have heart failure, including individuals with atrial fibrillation [3]. Traditional transvenous pacemakers (TPMs) have been the standard of care for decades, providing life-saving therapy to millions of patients worldwide [4]. However, the evolution of technology has driven the development of leadless pacemakers (LPMs) [5]. The first LPM was approved in Europe in 2012 and in the U.S. in 2016 [6]. These devices, which are entirely self-contained and implanted directly into the right ventricle, eliminate the need for leads and subcutaneous pockets, thereby reducing the infection rates and mechanical complications associated with traditional systems [7].

LPMs are particularly advantageous in patients with a high risk of infection, those who have experienced complications from leads, or those with venous obstructions that prevent the safe placement of leads [7]. Additionally, they offer a minimally invasive option for patients who may not be candidates for traditional pacemakers due to comorbid conditions [6]. Recent evidence from a large cohort study demonstrates a low prevalence of new-onset severe tricuspid regurgitation following LPM implantation [8]. Despite these benefits, LPMs are currently limited in terms of pacing modes (typically only providing single-chamber pacing), which restricts their use to specific patient populations, primarily those with atrioventricular block or sinus node dysfunction [9].

National-level data offer unique advantages, including the ability to capture population-specific trends and real-world outcomes that may not be fully represented in international registries or smaller-scale studies. Although previous systematic reviews have reported a low incidence of complications with LPM use [10], a synthesis of national-level data—capable of capturing population-specific trends and real-world outcomes—remains lacking. This study aims to address this gap by conducting a systematic review and meta-analysis comparing the acute complication and mortality rates associated with LPM versus TPM placements using data derived from national-level registries and administrative databases. The primary objective is to evaluate early safety outcomes by estimating pooled odds ratios (ORs) for acute overall complications, device-related complications, and mortality.

## 2. Materials and Methods

### 2.1. Search Strategy

To meet the objective of the present study, two outcome measures were used: acute mortality and acute overall complication and acute device-related complication rates. In-hospital and 30-day mortality, as reported by individual studies, were both classified under the term acute mortality, while in-hospital and 30-day complication rates were grouped under acute complications (overall and device-related) to harmonize reporting across studies. These measures were combined because they reflect early post-implantation outcomes and are commonly used to assess short-term procedural safety. A search in the PROSPERO database aimed to identify any registrations of comparable studies, but none were found. This study protocol is registered with the PROSPERO international prospective register of systematic reviews [10] (ID: CRD420251032513). Following this, a comprehensive search was conducted in the following electronic literature databases and search engines: PubMed, Scopus, ScienceDirect, and Google Scholar. The search included articles published up to 10 October 2024, with no restrictions on the depth of the review, as the first successful implantation of LPMs occurred in the 2010s. The search strategy incorporated the following keywords: “outcomes” OR “complications” AND “mortality” AND “leadless pacemaker” OR “Micra” OR “Nanostim”, with the following filters: English language and Humans.

### 2.2. Eligibility Criteria

The literature review and synthesis were conducted in alignment with the Preferred Reporting Items for Systematic Reviews and Meta-Analyses (PRISMA) guidelines, and the PRISMA 2020 checklist is available in the Appendix A [11]. Two independent researchers [A.S. (Akmoldir Sarsenbayeva) and K.B.] performed the search and screened the articles for relevance based on title and abstract. Articles were included if they specifically addressed the use of LPMs at a national level, providing clinical data on safety and outcomes. Database analysis, national claims data, and national registry analysis published in peer-reviewed journals were considered for inclusion. The full inclusion criteria were defined using the population, intervention, comparator, outcomes, and study design framework (PICOS) as follows: Population (P): adult patients who underwent single-chamber ventricular pacemaker implantation. Intervention (I): implantation of LPMs. Comparator (C): implantation of TPMs. Outcomes (O): reported data on acute complications (overall and device-related) and/or acute mortality, defined as in-hospital or 30-day outcomes. Study Design (S): studies based on national-level data (e.g., national databases or registries). Context: studies conducted in real-world, large-scale settings; only full-text articles published in English were considered.

Exclusion criteria included studies that did not provide a comparative analysis between LPM and TPM or presented irrelevant, incomplete, or duplicate data. Non-original research, such as systematic reviews, meta-analyses, editorials, commentaries, and abstracts, was excluded. Additionally, studies based on multinational registries, randomized clinical trials, or cross-sectional survey data rather than national-level data were not considered. In cases where multiple studies utilized the same national database, they were included in the analysis provided they did not present duplicative results. Discrepancies between researchers during the selection process were resolved through discussion and consultation with a third author [A.B.] until consensus was reached on all included studies.

### 2.3. Selection of Studies and Data Extraction

After identifying relevant publications, the duplicates were removed, and an initial screening was conducted by reviewing titles and abstracts. This was followed by a detailed eligibility assessment through full-text reviews, with exclusions made based on predefined criteria. In line with the PRISMA guidelines, two independent reviewers [A.S. (Akmoldir Sarsenbayeva) and K.B.] extracted key data from the full-text articles using a standardized extraction form. The extracted data were then compared and combined. Variables collected included the first author’s last name, year of publication, country of study, database used, study period, outcome assessment period, total number of patients receiving LPM and TPM placements, in-hospital or 30-day mortality, in-hospital or 30-day overall complication rates, in-hospital or 30-day device-related complication rates, types of complications, and the stated objective of each study.

### 2.4. Meta-Analysis Strategy

This study used R (version 4.3.2, released 31 October 2023) within the RStudio integrated development environment (version 2024.12.1+563) for statistical analysis [12]. Two R packages—meta and metafor—were used to perform the meta-analysis and calculate pooled ORs for acute mortality, acute overall complications, and device-related complication rates. Pooled ORs and their corresponding 95% confidence intervals (CIs) were calculated using a random-effects model and visualized using forest plots [13]. Heterogeneity across studies was assessed using the I^2^ statistic [14]. To evaluate the influence of individual studies—particularly those with larger sample sizes—an influence analysis was conducted. As publication bias assessment requires a minimum of 10 studies, funnel plots or Egger’s test analyses were not conducted [13]. Similarly, subgroup analysis was not performed due to the limited number of included studies.

### 2.5. Risk of Bias and Certainty of Evidence Assessment

The risk of bias (quality) of the included studies was assessed using the Critical Appraisal Skills Programme (CASP) cohort study checklist [15]. The checklist consisted of twelve questions, addressing aspects such as the study question, recruitment, exposure, outcome measure, confounding, follow-up, results of the study, applicability of the results, fit with other evidence, and implications of this study for practice. In ten out of twelve questions, each criterion received a rating of ‘yes’ (scored as 1) when adequately described, ‘no’ (scored as 0) when absent, and ‘can’t tell’ (scored as 0.5) when unclear or incomplete. The total scores ranged from 0 to 10, with a score of at least 7 indicating satisfactory quality.

The certainty of evidence was assessed using the Grading of Recommendations Assessment, Development, and Evaluation (GRADE) framework, following guidance from the Cochrane Handbook [16]. Domains evaluated included the risk of bias (via the CASP checklist), inconsistency (based on I^2^ thresholds: >75% = serious, >50% = moderate, >25% = not serious), and indirectness (judged against the PICOS criteria). Imprecision was considered by examining whether the 95% confidence interval of the pooled estimate could lead to different clinical interpretations [17]. Publication bias was not assessed [18].

## 3. Results

### 3.1. Included Study Characteristics

The initial database search yielded 1012 articles, of which 57 were excluded after applying the “Humans” and “English language” filters. Following the removal of 528 duplicates, 427 articles were screened for inclusion, and 96 were sought for retrieval. After full-text screening, five articles met the final inclusion criteria. These studies primarily examined the acute mortality of LPMs, acute overall complications associated with their use, and acute device-related complications compared to traditional TPMs.

Of the excluded articles, 41 did not include national-level data, 23 were review articles, and 10 did not report safety outcome data. The high number of review articles underscores the relevance and growing interest in this topic. Two studies were excluded for not reporting outcomes for both LPM and TPM groups [19,20], while two others reported only long-term outcomes (24 and 36 months) [21,22]. Four studies were excluded due to overlapping data [23,24,25,26], and four additional articles were excluded for other reasons, with references provided [27,28,29,30]. The PRISMA flow diagram detailing the study selection process is shown in Figure 1 [11].

All the included studies were published between 2021 and 2024. Four of the five studies analyzed were conducted in the United States using Medicare Claims Data, the Nationwide Readmission Database, and the National Inpatient Sample Database. One study was conducted in France using the Programme de Médicalisation des Systèmes d’Information (PMSI) database. In total, data on 28,470 LPM placements and 317,686 TPM placements were analyzed across the included studies. A summary of these studies is given below in Table 1.

### 3.2. LPM vs. TPM Placement Outcome Data

Table 2 summarizes the acute overall complications reported by the authors of the included studies. Clinical definitions of acute complications varied across studies. Of the five studies, only one did not report the rate of device-related complications during the early post-procedural period.

Figure 2A presents the pooled ORs for acute mortality comparing LPM (experimental) and TPM (control) recipients. The meta-analysis revealed no statistically significant difference in acute mortality between the two groups, with a pooled OR of 2.03 (95% CI: 0.65–6.34; I^2^ = 99%; *p* < 0.01), indicating that LPM implantation was not significantly associated with an increased risk of early mortality. Figure 2B displays the pooled ORs for acute overall complications, comparing LPM and TPM recipients. The meta-analysis showed a statistically non-significant increase in the odds of acute overall complications in the LPM group compared to the TPM group, with a pooled OR of 1.08 (95% CI: 0.45–2.61; I^2^ = 99%; *p* < 0.01). Similarly, Figure 2C illustrates the pooled ORs for acute device-related complications. The random-effects model indicated a statistically non-significant higher odds of device-related complications in the LPM group compared to the TPM group, with a pooled OR of 1.02 (95% CI: 0.23–4.44; I^2^ = 99%; *p* < 0.01).

Figure 3A displays the influence analysis of the pooled ORs for acute mortality comparing patients who received LPM versus TPM placement. The analysis indicates that the pooled OR was significantly influenced by the studies of Tonegawa-Kuji (2022) [33] and Bodin (2022) [32]. Figure 3B presents the influence analysis of the pooled ORs for acute overall complications comparing LPM and TPM recipients, showing a similar pattern of significant influence from the same two studies [32,33]. Figure 3C shows the influence analysis of the pooled ORs for acute device-related complications, revealing that the pooled estimate was notably influenced by the studies of Alhuarrat (2023) [34] and Crossley (2024) [35].

### 3.3. Risk of Bias and Certainty of Evidence Assessment

All included studies had a CASP score of 7 or above, indicating high quality and a low risk of bias, as shown in Table 3. None of the studies scored a 10, as it was difficult for the authors to determine whether all confounders had been accounted for in every included study.

Based on the GRADE certainty assessment results shown in Table 4, all three pooled estimates have a low certainty of evidence and should be interpreted with caution. Furthermore, in this meta-analysis, as only five study results were used in the meta-analysis, the publication bias assessment was not performed.

## 4. Discussion

In this study, a systematic review and meta-analysis of the literature was conducted to summarize the acute mortality and complication rates from national-level data on LPM versus TPM placements. The findings of this analysis offer interesting insights. During the analyzed period, TPM placements were more than 10 times more frequent than LPM placements. Although the analysis showed an increase in the odds of acute mortality, acute overall complications, and acute device-related complications in the LPM group compared to the TPM group, those differences were not statistically significant. These findings suggest comparable short-term safety profiles with respect to mortality, overall complications, and device-related complications across both pacing modalities. However, the low certainty of evidence regarding the meta-analysis results warrants cautious interpretation.

This finding supports the growing body of evidence suggesting a favourable short-term safety profile for LPMs in comparison to traditional transvenous systems. The results of the European Heart Rhythm Association (EHRA) survey indicate that local vascular complications were the most frequently reported issues in patients with LPM placements [36]. Additionally, more than one-third of the respondents of the EHRA survey reported that they had never encountered any complications after LPM placement in cardiac centres within the Electrophysiology Research Network. These findings further support the results of this analysis of short-term complications.

An existing meta-analysis of LPM placements, combining results from prospective and retrospective cohort studies, shows that LPM placement is associated with a low risk of complications up to one-year post-implantation [37]. While the present study reveals no difference in the early post-intervention period, national-level data in three studies show no difference in all-cause adjusted mortality during the longer follow-up period of 12 to 36 months [21,22,26]. Piccini and coauthors elaborate in their findings that in real-world national-level data analysis, patients are usually older and have more comorbidities than patients in registry trials [31]. This helps explain the absence of a difference in all-cause mortality between LPM and traditional pacemakers.

Beyond the early post-implantation period, recent large-scale national-level data have demonstrated sustained advantages of LPMs over TPMs. A Medicare-based analysis showed that LPM placements were associated with a 38% reduction in adjusted reintervention rates and a 31% reduction in adjusted chronic complication rates compared to TPM placements, with no significant difference in all-cause mortality at two years [21]. Similarly, a three-year follow-up of Medicare data confirmed lower risks of complications, infections, reinterventions, and heart failure hospitalizations among patients who received LPMs compared to TPMs, suggesting that the initial safety benefits of LPMs are maintained—and potentially enhanced—over time [22].

The findings of the present study also align with current guidelines that recommend leadless pacemakers, particularly for patients requiring single-chamber ventricular pacing, as a safer alternative to traditional systems [38]. The United Kingdom Expert Consensus (2022) [39] recommends the following patient groups for LPM placements: those at high risk of infection, patients with end-stage kidney disease, individuals with a history of device infection, those with anatomical constraints for transvenous pacemaker placement, patients using immunosuppressants or steroids, and other immunocompromised individuals. A database analysis from multiple centres in Europe reports a 98% success rate in leadless pacemaker placements, which aligns with other study findings [39]. El-Amrani and colleagues reported a 99.1% success rate in LPM placements in patients aged 90 years and above, supporting these analysis results [40]. However, it should be noted that the studies reviewed primarily focus on patients with indications for single-chamber pacing, limiting the generalizability to all pacemaker-dependent patients.

All of the included studies provided safety data on the single-chamber pacing capability of leadless devices. Single-chamber leadless pacemakers lack the capability to provide atrial pacing or maintain consistent atrioventricular synchrony. Recent data suggest that following three months of implantation, the dual-chamber leadless pacemaker met its safety benchmarks and ensured dependable atrial pacing along with stable atrioventricular synchrony [41]. Using dual-chamber leadless pacemakers may increase the number of patients who qualify for this procedure, although additional long-term studies are needed to confirm these results.

This study contains a few limitations. Detailed subgroup analyses by age, comorbidities, and device type were lacking, which might have influenced complication rates and patient outcomes. This study did not collect data on specific types of complications but rather aggregated complications into broader categories. This approach, while necessary due to the nature of the available data, limits the ability to analyze individual complication types. Additionally, all-cause mortality and all-cause complication data may not be directly attributable to the device or the LPM placement procedure itself, as these outcomes could be influenced by underlying comorbidities or intercurrent illnesses. The use of different national databases and methodologies could introduce bias into the overall interpretation. The low certainty of evidence and high heterogeneity across the included studies underscore the need for cautious interpretation. Furthermore, the geographic variability of the included studies is limited to high-income countries. Conducting a comprehensive analysis of national-level data from various geographic regions and income settings would enhance the generalizability of these findings.

## 5. Conclusions

This study is the first systematic review and meta-analysis to synthesize national-level data comparing acute safety outcomes of LPM and TPM placements. The findings demonstrate that LPM placement is not significantly associated with increased acute mortality, overall complications, or device-related complications compared to TPM, suggesting a comparable short-term safety profile across both modalities. These results reinforce the clinical viability of LPMs as a safe alternative to traditional systems, particularly for patients requiring single-chamber pacing. Future technological advancements may further expand their applicability. Additional studies involving more diverse patient populations and extended follow-up periods are needed to support broader clinical recommendations for the use of LPMs in practice.

## Figures and Tables

**Figure 1 medicina-61-00974-f001:**
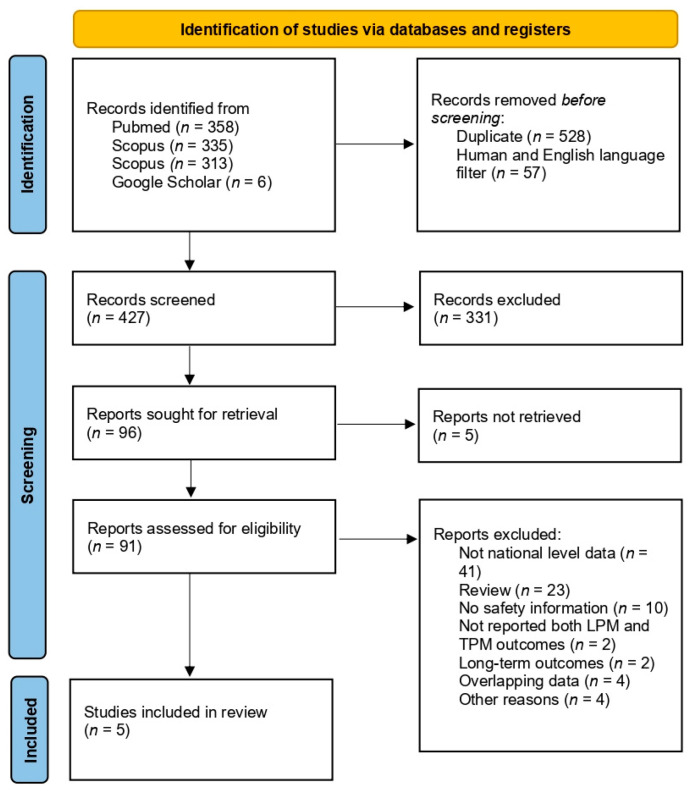
Flowchart of study screening and inclusion based on PRISMA guidelines.

**Figure 2 medicina-61-00974-f002:**
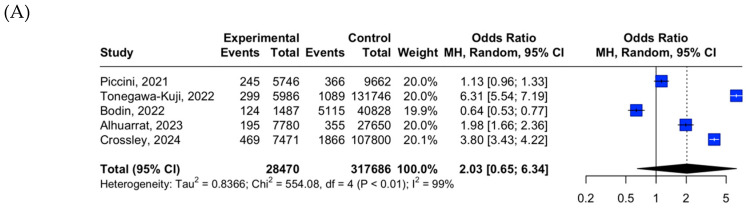
Meta-analysis of LPM versus TPM placement outcomes: (**A**) acute mortality; (**B**) acute overall complications; (**C**) acute device-related complications. Studies included to the meta-analysis: Piccini, 2021 [31], Bodin, 2022 [32], Tonegawa-Kuji, 2022 [33], Alhuarrat, 2023 [34], Crossley, 2024 [35].

**Figure 3 medicina-61-00974-f003:**
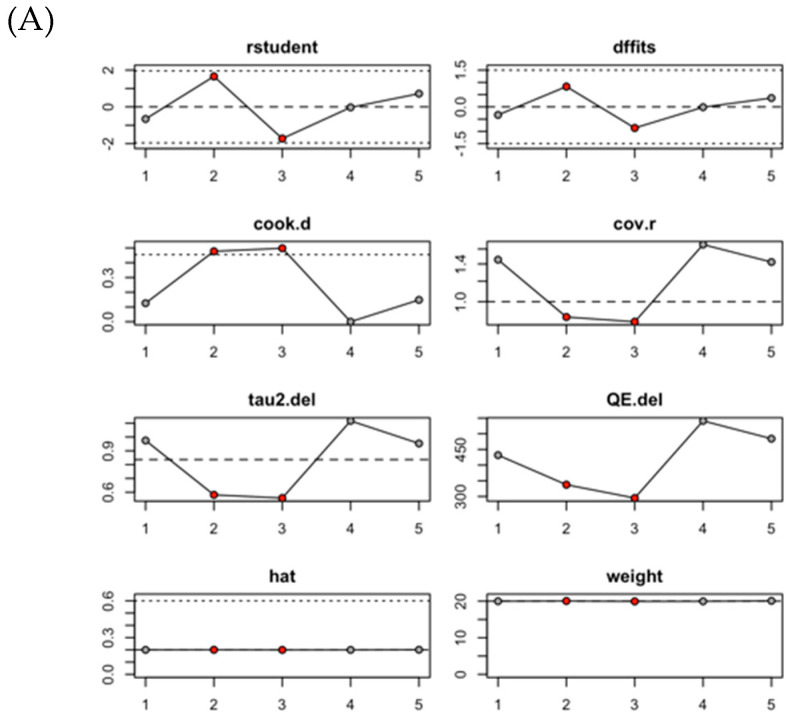
Influence analysis of pooled ORs for LPM versus TPM outcomes: (**A**) acute mortality; (**B**) acute overall complications; (**C**) acute device-related complications.

**Table 1 medicina-61-00974-t001:** Leadless pacemaker placement studies.

Study	Country	Database Name	Period	Sample Size		Age Mean ± SD or Median (IQR)		Objective
				LPM	TPM	LPM	TPM	
Piccini, 2021 [31]	USA	Medicare Claims Data	9 March 2017–1 December 2018	3726	7265	79.4 ± 9.5	82.0 ± 8.1	30-day and 6-month complication rates
Bodin, 2022 [32]	France	PMSI	1 January 2017–1 September 2020	1487	40,828	70.7 ± 18.4	83.4 ± 9.1	30-day complication rate, including all-cause death and cardiovascular death
Tonegawa-Kuji, 2022 [33]	USA	Nationwide Readmissions Database	April 2017–December 2019	5986	131,746	79 (70–86)	78 (70–85)	In-hospital complications and 30-day readmission
Alhuarrat, 2023 [34]	USA	National Inpatient Sample Database	2016–2019	7780	27,650	77.1 ± 12.1	81.3 ± 9.4	In-hospital outcome and procedural complications
Crossley, 2024 [35]	USA	Medicare Claims Data	5 February 2020–1 December 2021	7471	107,800	79.0 ± 10.2	78.7 ± 8.0	30 days and 6 months overall and device-related complications, and all-cause mortality

Abbreviations: IQR—interquartile range; LPM—leadless pacemaker; PMSI—Programme de Médicalisation des Systèmes d’Information; SD—standard deviation; TPM—transvenous pacemakers; USA—United States of America.

**Table 2 medicina-61-00974-t002:** Acute complications reported by the authors of the included studies.

Study	Complication Assessment Period	Acute Complications
Piccini, 2021 [31]	30-day	Embolism and thrombosis, puncture site events, cardiac effusion and perforation, device-related, and other complications.
Bodin, 2022 [32]	In-hospital	Embolism and thrombosis, puncture site events, cardiac effusion and perforation, device-related, and other complications.
Tonegawa-Kuji, 2022 [33]	30-day	Cardiac tamponade, pneumothorax, hemothorax, major bleeding, transfusion, and device-related complications.
Alhuarrat, 2023 [34]	In-hospital	Bleeding, vascular events, venous thromboembolism, device-related, cardiac, pulmonary, infection, and neurologic complications.
Crossley, 2024 [35]	30-day	Embolism and thrombosis, puncture site events, cardiac effusion and perforation, device-related, and other complications.

**Table 3 medicina-61-00974-t003:** The risk of bias assessment results from the Critical Appraisal Skills Programme (CASP) checklist.

Authors	Question	Recruitment	Exposure	Outcome	Confounding	Follow up	Results	Applicability	Fit	Implications	Score
Piccini, 2021 [31]	Yes	Yes	Yes	Yes	Cannot t tell	Yes	Yes	Yes	Cannot tell	Yes	9.0
Bodin, 2022 [32]	Yes	Yes	Yes	Yes	Cannot tell	Yes	Yes	Yes	Cannot tell	Yes	9.0
Tonegawa-Kuji, 2022 [33]	Yes	Yes	Yes	Yes	Cannot tell	Yes	Yes	Cannot tell	Cannot tell	Yes	8.5
Alhuarrat, 2023 [34]	Yes	Yes	Yes	Yes	Cannot tell	Can’t tell	Yes	Yes	Cannot tell	Yes	8.5
Crossley, 2024 [35]	Yes	Yes	Yes	Yes	Cannot tell	Yes	Yes	Yes	Cannot tell	Yes	9.0

**Table 4 medicina-61-00974-t004:** GRADE certainty of evidence assessment.

Outcome	Risk of Bias	Inconsistency	Indirectness	Imprecision	Publication Bias	Certainty of Evidence
Acute mortality	Low	Serious	Not serious	Serious	Not assessed	Low
Acute overall complications	Low	Serious	Not serious	Serious	Not assessed	Low
Acute device-related complications	Low	Serious	Not serious	Serious	Not assessed	Low

## Data Availability

The original contributions presented in this study are included in this article. Further inquiries can be directed to the corresponding author.

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
