# Peer review of "Systematic Review and Meta-Analysis of Acute Mortality and Complication Rates Following Leadless Pacemaker Placement Using National-Level Data"

_medicina, 2025, doi:10.3390/medicina61060974_

Round 1
Reviewer 1 Report
Comments and Suggestions for Authors
Congratulations to the authors for having produced a manuscript about such a relevant topic as leadless pacing; I just have some comments about it:
While the authors mention that heterogeneity was evaluated using the I² statistic, they do not report the specific I² values corresponding to the three analyzed outcomes. Including these figures would enhance transparency and aid in interpreting the consistency of the findings across studies.
Given the rapid pace of advancements in this field, the review would be enriched by broader contextual insights. In particular, the development of dual-chamber leadless pacemakers—such as the Micra AV and AVEIR DR—marks a pivotal shift in pacing technology. As demonstrated by Knops et al. in NEJM (2023), these innovations begin to overcome the historical constraints of LPMs being limited to single-chamber use. Even a brief discussion of these devices would underscore the expanding clinical scope of leadless pacing in the near future.
Authors must include in their discussion a paragraph about the potential impact of leadless pacemakers on tricuspid regurgitation and the most recent evidences (doi: 10.1016/j.hrthm.2024.06.030).
Although this review appropriately concentrates on short-term outcomes, several of the excluded studies explored long-term endpoints such as mortality and complications over 12 to 36 months. Including a brief comparison between these long-term findings and the short-term results synthesized here would provide readers with a more comprehensive view of the evidence landscape.
Funnel plots and a GRADE summary of findings table are notably absent from the manuscript, authors should provide them.
Author Response
We thank the reviewer for their invaluable feedback, which has contributed to enhancing the quality and clarity of our manuscript for its potential readers. Our responses are in the attached file.

Reviewer 2 Report
Comments and Suggestions for Authors
The meta-analysis compared the rates of acute complications and mortality associated with leadless pacemakers versus transvenous pacemaker placements using national-level data. Although it is well-organized, some modifications are necessary.
- The rationale for the study should be explained more clearly. Please incorporate references to available data and emphasize the existing gaps in the literature.
- The duration of the search must be included in the abstract.
- In the abstract, please include the P-value of the outcome. Please follow these directions in the results section of the manuscript and include the P-value.
- It is suggested that all instances of "we" or “our” be replaced with phrases such as “current study," "this study," or "present study".
- Some paragraphs are excessively long. If a paragraph exceeds 200 to 300 words, it is likely too lengthy for most readers to comprehend easily. Breaking it into smaller sub-paragraphs can significantly enhance readability. For example, consider the first paragraph of the introduction and the eligibility criteria section.
- The historical development of pacemakers should be removed from the introduction and replaced with relevant data about the objectives of this study.
- The resolution of the figures is inadequate.
- The conclusion should be revised to highlight the key outcomes that address the main question of the study.
- The quality of evidence based on GRADE and the risk of bias (RoB) status of the studies should be mentioned in the section on strengths and limitations.
Extensive proofreading is required.
Author Response

(The authors gave the same response as above.)

Round 2
Reviewer 1 Report
Comments and Suggestions for Authors
you have appropriately answered to all of my comments in the right way.
Reviewer 2 Report
Comments and Suggestions for Authors
The authors tried to address major comments. The manuscript has undergone revisions and has potential for publication.
Please remove "we" from the abstract.
In addition, the p-value associated with the confidence interval was requested for the abstract instead of the one related to heterogeneity.
Comments on the Quality of English LanguageMinor proofreading is required.